# The Prevalence, Management and Impact of Dysmenorrhea on Medical Students’ Lives—A Multicenter Study

**DOI:** 10.3390/healthcare10010157

**Published:** 2022-01-14

**Authors:** Romina-Marina Sima, Mihaela Sulea, Julia Caroline Radosa, Sebastian Findeklee, Bashar Haj Hamoud, Mihai Popescu, Gabriel Petre Gorecki, Anca Bobircă, Florin Bobirca, Catalin Cirstoveanu, Liana Ples

**Affiliations:** 1Department of Obstetrics and Ginecology, “Carol Davila” University of Medicine and Pharmacy, 020021 Bucharest, Romania; romina.sima@umfcd.ro (R.-M.S.); mihaelasulea25@gmail.com (M.S.); liana.ples@umfcd.ro (L.P.); 2The “Bucur” Maternity, “Saint John” Hospital, 040294 Bucharest, Romania; gabriel.gorecki@utm.doc.ro; 3Department for Gynaecology, Obstetrics and Reproductive Medicine, Saarland University Hospital, Kirrberger Straße 100, Building 9, 66421 Homburg, Germany; julia.radosa@uks.eu (J.C.R.); sebastian.findeklee@arcor.de (S.F.); bashar.hajhamoud@uks.eu (B.H.H.); 4Department of Anaesthesia and Critical Care, “Carol Davila” University of Medicine and Pharmacy, 020021 Bucharest, Romania; mihai.popescu@umfcd.ro; 5Faculty of Medicine, Titu Maiorescu University, 040441 Bucharest, Romania; 6Department of Internal Medicine and Rheumatology, “Carol Davila” University of Medicine and Pharmacy, 050474 Bucharest, Romania; 7Department of Surgery, “Carol Davila” University of Medicine and Pharmacy, 050474 Bucharest, Romania; florin.bobirca@umfcd.ro; 8Department of Pediatrics, “Carol Davila” University of Medicine and Pharmacy, 020021 Bucharest, Romania; catalin.cirstoveanu@umfcd.ro; 9Pediatrics Department, ‘Maria Sklodowska Curie’ Emergency Children Clinical Hospital, 041451 Bucharest, Romania

**Keywords:** dysmenorrhea, students, quality of life, pain intensity, menstruation

## Abstract

Introduction: Dysmenorrhea is defined as the presence of painful menstruation, and it affects daily activities in different ways. The aims of this study were to assess the prevalence and management of dysmenorrhea and to determine the impact of dysmenorrhea on the quality of life of medical students. Material and methods: The study conducted was prospective, analytical and observational and was performed between 7 November 2019 and 30 January 2020 in five university centers from Romania. The data was collected using an original questionnaire regarding menstrual cycles and dysmenorrhea. The information about relationships with family or friends, couples’ relationships and university activity helped to assess the effects of dysmenorrhea on quality of life. The level of significance was set at *p* < 0.05. Results: The study comprised 1720 students in total. The prevalence of dysmenorrhea was 78.4%. During their menstrual period, most female students felt more agitated or nervous (72.7%), more tired (66.9%), as if they had less energy for daily activities (75.9%) and highly stressed (57.9%), with a normal diet being difficult to achieve (30.0%). University courses (49.4%), social life (34.5%), couples’ relationships (29.6%), as well as relationships with family (21.4%) and friends (15.4%) were also affected, depending on the duration and intensity of the pain. Conclusion: Dysmenorrhea has a high prevalence among medical students and could affect the quality of life of students in several ways. During their menstrual period, most female students feel as if they have less energy for daily activities and exhibit a higher level of stress. The intensity of the symptoms varies considerably and, with it, the degree of discomfort it creates. Most student use both pharmacological and non-pharmacological methods to reduce pain (75.7%). University courses, social life, couples’ relationships, as well as relationships with family and friends are affected, depending on the duration and intensity of the pain.

## 1. Introduction

The term dysmenorrhea originates from Greek and can be translated as “abnormal monthly flow” [1,2,3]. Dysmenorrhea is a common condition among women of reproductive age and can affect up to 90% of them [4,5,6]. It can be divided into two categories: primary dysmenorrhea and secondary dysmenorrhea [2,6,7,8]. The pain is characterized by muscle cramps [2,9]. It is also often accompanied with symptoms such as nausea, vomiting, diarrhea, lower-back pain and headaches [10,11,12].

Over time, numerous theories have emerged that have attempted to identify the cause of primary dysmenorrhea. Hippocrates explained that painful menstruation occurs as a result of obstruction of the cervical canal, which causes a stagnation of blood in the uterus. The theory is supported by the fact that nulliparous women have more intense pain than multiparous women [1]. Primary dysmenorrhea has been associated with myometrial hyperactivity. Increased pressure in the uterine cavity has been considered a possible cause of menstrual pain [13]. This is followed by decreased uterine blood flow, thus leading to tissue ischemia and pain [14]. Primary dysmenorrhea is associated with the presence of ovulatory menstrual cycles [1,2,15]. Thus, immediately after menarche, painless menstruation may be present because ovulation does not occur [1].

Secondary dysmenorrhea occurs due to the existence of a pelvic pathology. The most common cause is endometriosis [4]. Secondary dysmenorrhea usually occurs a few years after menarche [1,16,17]. The pain may be localized or diffuse in the lower abdomen and is not necessarily limited to the menstrual period. Pain may be accompanied by other symptoms, such as menorrhagia or intermenstrual bleeding [17]. Abdominal discomfort may be associated with defecation, sexual contact, fever or abnormal bleeding [16]. Secondary dysmenorrhea can have many causes, such as adenomyosis, uterine fibroids, endometrial polyps, ovarian cysts, cervical canal stenosis, retroverted uterus or the use of intrauterine devices [1].

Dysmenorrhea has a negative effect on the quality of life of patients because it affects relationships with family and friends, university (or professional) performance, as well as leisure activities [17,18,19,20]. Regarding university activity and the individual’s studies, the ability to concentrate on the courses and the volume of information accumulated are influenced by the presence of the pain [18,20]. The negative impact on academic performance was proved for both school and higher education in a meta-analysis [21]. Daily activities are affected in different ways. When they were menstruating, 38.4% of women included in the study by Schoep et al. stated that they carried out fewer activities or were unable to do anything [22]. There are also women who avoid exercising during menstruation; 50% of those studied by Vlachou et al. [18]. During menstruation, the quality of sleep declines, compared to other periods of the menstrual cycle [17]. Chronic pelvic pain among women and dysmenorrhea is frequently underestimated [23]. Dysmenorrhea is associated with increased anxiety and depression [18] and general practitioners need to be ready to discuss these issues with patients [24]. There are women who say that severe menstrual symptoms increase the risk of exposure to high doses of drugs while they try to reduce the discomfort [25].

The aim of this study was to determine the impact of dysmenorrhea on the quality of life of medical students. Other objectives were to evaluate the prevalence and management of dysmenorrhea among students, as well as the evaluation of menstrual pain and its effect on daily activities.

## 2. Materials and Methods

The study we conducted is prospective, analytical, observational and took place between 7 November 2019 and 30 January 2020. The study population included female medical students from five university centers in the country of Romania: Bucharest, Cluj-Napoca, Craiova, Târgu Mureș and Timișoara.

We analyzed several published articles regarding the impact of dysmenorrhea on students’ quality of life and we developed a questionnaire that was used to achieve the objectives of this study (Appendix A). The data were collected using our original questionnaire of 59 questions grouped into three sections, according to the evaluation parameters: general characteristics, menstrual pain and the impact of dysmenorrhea on quality of life. We posed 52 closed questions, with two or more answer options, of which 12 were multiple-choice questions. In addition, seven were open-ended questions. Some of the questions were similar to those included in previously published articles [18,19,20,26,27]. The students who stated that they do not have dysmenorrhea answered only the first section of questions.

The questionnaire helped to collect data about the medical history and lifestyles of the students. Various features of menstrual cycles (regularity, duration of menstruation, degree of bleeding, premenstrual syndrome) and dysmenorrhea (onset, duration, location and intensity of pain, associated symptoms, methods used to reduce pain) have been evaluated. The impact of menstrual pain on the quality of life of students was monitored by collecting information about relationships with family members or friends and couples’ relationships, but also about university performance. The questionnaire was constructed using the online tool Google Forms and was afterwards distributed through social media to medical student groups.

### Statistical Analyses

The information collected was entered into a database created using the SPSS Statistic software, version 23 (Armonk, NY, USA).

Using descriptive statistical methods, we calculated frequencies and percentages, means and standard deviations, corresponding to each type of variable. Bar charts were created to this end. Statistical analysis was performed using the following tests: Chi-square, Independent-Samples t Test. Binary logistic regression was performed to determine the odds ratio (OR) and the confidence interval (95% Cl). The test results were considered statistically significant when the p value provided by them had a value of less than 0.05 (*p* < 0.05).

## 3. Results

This study included a 1720 Romanian medical students from five university centers throughout the country. The highest percentage of participants was from Bucharest (58.7%—1010 students).

The students were aged between 18 and 30. The average age of the participants was 22.06 ± 2.01 years. The age of the first menstruation varied between 8 and 18 years and its average was 12.39 ± 1.33 years. Most participants (1312 students—76.3%) reported regular menstrual cycles (that last between 21 and 35 days) (Table 1).

Out of 1720 participants, 1349 (78.4%) had dysmenorrhea and answered questions about its characteristics, but also about the methods used to relieve pain.

For 594 students (44.0%), dysmenorrhea appeared from the first menstruation. The majority of the students reported that menstrual pain began on the first day of menstruation (60.9%). Dysmenorrhea was present at each menstruation in 615 cases (45.6%). A total of 1284 participants (95.2%) reported that pain was localized to the pelvis and lower abdomen. In most cases the pain radiated only to the lumbar region (40.5%). The pain persisted for, at most, an entire day in the case of 572 students (42.4%).

Altogether 778 students (57.7%) considered their menstrual pain to be moderate, 507 students (37.6%) considered their menstrual pain to be severe, and only 4.7% of students said that the pain was mild. Asking if they thought the pain was more intense when they had a stressful period, 51.7% of participants answered affirmatively. After becoming sexually active, 124 students (9.2%) said that the intensity of pain had decreased, but 950 students (70.4%) did not agree with this.

The students were asked to assess the severity of the pain, assigning it an intensity number ranging from 1 to 10, with 1 representing the absence of pain and 10 representing an unbearable pain. The mean pain intensity was 6.99 ± 1.62.

Using the Independent-Samples t Test, we found that there was no statistically significant difference in pain intensity (rated from 1 to 10) with an active or inactive sex life (*p* = 0.360), with or without the use of oral contraceptives (*p* = 0.818).

The students were asked about the methods used to relieve pain. We observed that 1021 students (75.7%) used both pharmacological and non-pharmacological methods to reduce pain, 234 students (17.3%) used only pharmacological methods, while 65 students (4.8%) preferred only non-pharmacological methods. The most widely used drugs were non-steroidal anti-inflammatory drugs, utilized by 741 students (54.9%). Antispasmodics were used by 131 students (9.7%). Taking a combination of non-steroidal anti-inflammatory drugs and antispasmodics was an option for 336 students (24.9%). The timing for starting medication differed from student to student. Medications were taken when the pain became unbearable for 839 students (62.2%). Regardless of the intensity of the pain, its occurrence was an impulse for the administration of drugs for 421 students (31.2%). It is noteworthy that 131 students (9.7%) started taking medication before the onset of pain only when they had scheduled events or activities. Some students (5.9%) started taking medication every time, before the onset of pain. Medication could help to perform daily activities under normal conditions. Most students agreed with this statement (81.2%).

Non-pharmacological methods were used by a significant number of students to relieve pain (Table 2).

Pain was not the only symptom that could occur during menstruation. It might be accompanied by a number of other symptoms, more or less stressful (Table 3). Agitation or irritability were considered by 348 students (25.8%) as the most unpleasant symptoms that could accompany pain.

Dysmenorrhea affected the quality of life of 868 students (64.3%), while 481 students (35.7%) did not agree.

Most students (81.4%) considered that the intensity of pain was its most important characteristic, responsible for affecting their quality of life.

Regarding holidays or excursions, 619 students (45.9%) said that they planned such events according to the menstrual period in order to avoid the presence of pain during moments of relaxation. It was also important that 990 students (73.4%) changed their clothing style when they were menstruating and preferred to wear pants.

Dysmenorrhea could affect quality of life in many ways (Table 4).

Dysmenorrhea had a negative effect on university activities for 666 students (49.4%). Menstrual cramps affected social life for 466 students (34.5%). If dysmenorrhea was present, couples’ relationships suffered in the case of 399 students (29.6%). Relationships with family members and relationships with friends could also be affected, according to 289 students (21.4%) and 208 students (15.4%), respectively. The distribution of students according to the impact on their quality of life and the duration or intensity of pain is presented in Table 5.

Students with menstrual pain lasting 2 or more days were more likely to be affected in terms of university activity (OR = 2.21, 95% Cl = 1.65–2.95), social life (OR = 2.01, 95% Cl = 1.49–2.73), relationships with family members (OR = 1.66, 95% Cl = 1.17–2.35), couples’ relationships (OR = 1.78, 95% Cl = 1.31–2.43) or relationships with friends (OR = 1.63, 95% Cl = 1.10–2.41) compared to students whose pain persisted for several hours.

Participants who said that their menstrual pain was severe were more likely to observe an effect on relationships with friends (OR = 5.90, 95% Cl = 1.82–19.15), university activity (OR = 13.63, 95% Cl = 6.36–29.25), couples’ relationships (OR = 6.95, 95% Cl = 2.74–17.64), relationships with family members (OR = 13.52, 95% Cl = 3.27–55.99) or social life (OR = 14.48, 95% Cl = 5.18–40.43) compared to students who had mild menstrual pain.

Regarding family relationships, 506 students (37.5%) considered that during the menstrual period they had conflicts with family members due to the condition related to the presence of dysmenorrhea. A total of 326 students tried to isolate themselves from family members when dysmenorrhea occurred.

In the case of 418 students (31.0%), the presence of dysmenorrhea determined the avoidance of meetings with friends, while in the case of 309 students (22.9%) it led to fewer discussions with friends.

Couples’ activities were occasionally affected by dysmenorrhea among 603 students (44.7%) and were not affected by it for 552 students (40.9%). When they were menstruating, 695 students (51.5%) considered that they received increased attention from their life partner.

University activities were affected in 49.4% of students. During the menstrual period, some students did not have to miss classes, but 461 students (34.2%) usually missed a day of classes. Approximately 4.8% of students missed 2 or more days, of which 64 students (4.7%) missed 2–3 days, and one student (0.1%) missed more than 3 days. Dysmenorrhea could influence university performance by affecting various aspects. Thus, 808 students (59.9%) claimed that they failed to focus on courses. Individual study was affected in 782 students (58.8%). On the other hand, 569 students (42.2%) considered that they failed to accumulate the same amount of information when dysmenorrhea was present. Some of the students missed exams due to dysmenorrhea (4.2%) or had the opinion that the condition influenced an exam grade (18.4%).

The effect of dysmenorrhea on university performance depending on the duration and intensity of pain was analyzed (Table 6). If the pain persisted for 2 days or more, students were more likely to be unable to concentrate on classes (OR = 1.92, 95% Cl = 1.43–2.58), the volume of information would be smaller (OR = 2.16, 95% Cl = 1.61–2.90) and the individual studies were more likely to be affected (OR = 2.13, 95% Cl = 1.60–2.85) than if the pain occurred for only a few hours.

Students who had severe pain were more likely to be unable to focus on classes (OR = 10.15, 95% Cl = 5.43–18.96), have their individual studies impacted (OR = 4.90, 95% Cl = 2.80–8.60) or be unable to accumulate the same amount of information (OR = 8.99, 95% Cl = 4.20–19.24) as those with mild pain.

## 4. Discussion

The number of participants from our study was higher than in other studies that have assessed the impact of dysmenorrhea on quality of life and in which 637 students, [18] 623 students, [28] 956 students, [19] or 1092 participants were included [20]. The prevalence of dysmenorrhea among medical students was 78.4%, which is similar to the results presented in other studies: 78.8% [20] and 72.7% [28]. A prevalence of dysmenorrhea, ranging from 16% to 91%, has been reported in the literature [8,18,25].

The average age of the students included in the study was 22.06 ± 2.01 years and the average age for the first menstruation was 12.39 ± 1.32 years. A similar age of menarche has been reported: 12.74 ± 1.57 years, [18] 12.3 ± 1.3 years [29] and 12.6 ± 1.04 years [30].

For most students the pain began on the first day of menstruation and was located mainly in the pelvis and lower abdomen; this data corresponded to the data presented in the literature [20,31].

Sahin et al. found approximately equal proportions for pain intensity: mild (36.2%), moderate (34.3%) and severe (29.5%) [32]. After processing the students’ answers, we found that menstrual pain in 57.7% of cases was moderate, in 37.7% of cases it was severe and that only 4.7% of students had mild pain. A significant proportion of students with moderate to severe pain have been reported in other studies [18,19]. The mean severity of pain among medical students was 6.99 ± 1.61, similar to that reported by Helwa et al., 6.79 ± 2.64 [19].

The methods used by participants to relieve pain were evaluated in several studies. Out of the total medical students included in this study, 75.7% used both pharmacological and non-pharmacological methods to relieve pain. Helwa et al. found that 41.5% of students did not use medication to reduce menstrual pain [19] Nonsteroidal anti-inflammatory drugs (48.3%) and Paracetamol (43.2%) were preferred by students included in the study by Vlachou et al. [18] Spasmolytic drugs were mostly used by students in the study by Mohamed et al. (71.8%) [20]. Non-steroidal anti-inflammatory drugs were used by 54.9% of the students included in this study. Spasmolytics were used by 9.7% of students, while 24.9% of students used both categories of drugs.

The most widely used non-pharmacological methods were the application of liquids or hot objects on the abdomen, sleep, massaging of painful regions and the consumption of sweets. Similar results about non-pharmacological methods have been reported in the literature [20].

In addition to pain, menstruation could cause other symptoms that were more or less stressful and that could also affect the quality of life of students. In this study, we found that the most common symptoms associated with dysmenorrhea were: agitation or irritability, fatigue, headache, diarrhea, nausea and experiencing dizziness. These symptoms were also found to be among the most common in the participants included in other studies [19,21]. Arthralgia was found in a significant number of students (28.6%) in the study published by Helwa et al., while we found that only 9.0% of students also had this symptom during their menstrual period [19]. Loss of appetite for food was the most common symptom (51.9%) associated with menstrual pain among participants in the study by Mohamed et al. [20].

We found that 63.4% of medical students believe that dysmenorrhea affected their quality of life. Among the characteristics of pain, its intensity was the most incriminated for the effect of dysmenorrhea on quality of life. The intensity of the pain was significantly higher among students who said so. The most analyzed aspects of life that could be affected by dysmenorrhea were university performance, relationship with family and relationships with friends [18,19,20]. In addition to these, we included questions about couples’ relationships in the questionnaire. Of all of the above, we found that university activity was affected in 49.4% of students, followed by social life (34.5%) and couples’ relationships (29.6%). At the opposite end of the spectrum are relationships with friends (15.4%). Relationships with family members were affected in 21.4% of students. Mohamed et al. reported that 78.3% of their participants believed that family relationships suffered from dysmenorrhea. This difference between the two results may be due to the fact that the average age of the participants in the study conducted by Mohamed et al. was 16.8 ± 0.876 years, lower than the average age of the students in this study (22.06 ± 2.01 years); thus, the interaction with the family would be different depending on age [20]. In short, during the menstrual period, a greater number of conflicts may have occurred, students might have isolated themselves from family members or they may have been less concerned with family duties.

The duration and intensity of pain had a significant effect on university performance, relationships with family members and friends and on couples’ relationships.

Regarding university activity, the individual studying, the ability to concentrate on the courses and the volume of information accumulated were significantly influenced by the intensity of the pain, the results being similar to those presented by Vlachou et al. [18]. In addition, we analyzed the impact of the duration of dysmenorrhea on these situations and concluded that they were significantly influenced by the duration of the presence of pain. Due to dysmenorrhea, 39.0% of students missed classes, with most of them (34.2%) having to skip a day. Similar data are presented by Mohamed et al. [20].

This study has limitations due to the subjectivity and truthfulness of the answers given by students, but also due to the fact that most of the questions were closed in nature, not allowing participants to customize the answers.

## 5. Conclusions

Dysmenorrhea has a high prevalence among medical students and could affect the quality of life of students in several ways. During their menstrual period, most female students feel as if they have less energy for daily activities and exhibit a higher level of stress. The intensity of the symptoms varies a lot and, with it, the degree of discomfort it creates. Most student use both pharmacological and non-pharmacological methods to reduce pain (75.7%). The most widely used drugs are non-steroidal anti-inflammatory drugs (54.9%). Non-pharmacological methods are used by a significant number of students to relieve pain. University courses, social life, couples’ relationships, as well as relationships with family and friends are affected, depending on the duration and intensity of the pain.

## Figures and Tables

**Table 1 healthcare-10-00157-t001:** Characteristics of Female Students (*n* = 1720).

Characteristics	Dysmenorrhea (%)		Total (%)	χ^2^	*p*
	No	Yes			
Menstrual Regularity				1.898	0.168
Irregular	98 (26.4)	310 (23.0)	408 (23.7)		
Regular	273 (73.6)	1039 (77.0)	1312 (76.3)		
Duration of menstruation				24.644	0.000
1–3 days	42 (11.3)	101 (7.5)	143 (8.3)		
3–5 days	240 (64.7)	745 (55.2)	985 (57.3)		
>5 days	89 (24.0)	503 (37.3)	592 (34.4)		
Degree of bleeding				39.390	0.000
Normal bleeding	241 (65.0)	817 (60.6)	1058 (61.5)		
Heavy bleeding	70 (18.9)	428 (31.7)	498 (29.0)		
Less bleeding	60 (16.2)	104 (7.7)	164 (9.5)		
Premenstrual syndrome				141.497	0.000
Yes, at each menstrual cycle	175 (47.2)	1021 (75.7)	1196 (69.5)		
Occasional	149 (40.2)	298 (22.1)	447 (26.0)		
No	47 (12.7)	30 (2.2)	77 (4.5)		
Gynecological surgeries				0.037	0.847
Yes	12 (3.2)	41 (3.0)	63 (3.1)		
No	359 (96.8)	1308 (97.0)	1667 (96.9)		
Autoimmune disease				1.737	0.188
Yes	9 (2.4)	52 (3.9)	61 (3.5)		
No	362 (97.6)	1297 (96.1)	1659 (96.5)		
Family history of dysmenorrhea				230.458	0.000
Yes	106 (28.6)	967 (71.7)	1073 (62.4)		
No	265 (71.4)	382 (28.3)	647 (37.6)		
Regular physical activity				5.847	0.016
Yes	194 (52.3)	610 (45.2)	804 (46.7)		
No	177 (47.7)	739 (54.8)	916 (53.3)		
Healthy nutrition				4.314	0.038
Yes	203 (54.7)	656 (48.6)	859 (49.9)		
No	168 (45.3)	693 (51.4)	861 (50.1)		
Smoking				0.582	0.748
Yes	100 (27.0)	375 (27.8)	475 (27.6)		
Former smoker	54 (14.5)	213 (15.8)	267 (15.5)		
Never	217 (58.5)	761 (56.4)	978 (56.9)		
Sleeping (hours)				4.153	0.125
<6	59 (15.9)	274 (20.3)	333 (19.4)		
6–9	308 (83.0)	1066 (79.0)	1374 (79.8)		
>9	4 (1.1)	9 (0.7)	13 (0.8)		
Sex life				13.343	0.000
Active	233 (62.8)	979 (72.6)	1212 (70.5)		
Inactive	138 (37.2)	370 (27.4)	508 (29.5)		

**Table 2 healthcare-10-00157-t002:** Non-pharmacological Methods (*n* = 1349).

Non-Pharmacological Method	Number of Participants	Percent (%)
Applying warm liquids or objects to the abdomen	737	54.6
Sleeping	689	51.1
Massaging painful regions	490	36.3
Eating sweets	265	19.6
Walking	130	9.6
Other methods	53	3.9
Warm pads	27	2.0

**Table 3 healthcare-10-00157-t003:** Symptoms Associated with Dysmenorrhea (*n* = 1349).

Symptoms	Number of Participants	Percent (%)
Agitation or irritability	902	66.9
Fatigue	880	65.2
Headache	625	46.3
Diarrhea	607	45.0
Nausea	437	32.4
Dizziness	413	30.6
Loss of appetite	378	28.0
Sweating	378	28.0
Polyuria	237	17.6
Insomnia	150	11.1
Vomiting	131	9.7
Arthralgia	121	9.0
Only pain	73	5.4

**Table 4 healthcare-10-00157-t004:** Dysmenorrhea and Quality of Life (*n* = 1349).

Negative Effects	Number of Participants	Percent (%)
Decreasing in energy levels for daily activities	1024	75.9
Feeling more agitated or nervous	981	72.7
Feeling more tired	902	66.9
Having higher levels of stress	781	57.9
Not performing normal physical activities	625	46.3
Not having a normal diet	405	30.0

**Table 5 healthcare-10-00157-t005:** Quality of Life and Duration or Intensity of Pain (*n* = 1349).

Variable	Duration of Pain (%)			Intensity of Pain (%)		
	Few Hours	One Day	Two Days or More	Mild	Moderate	Severe
**University activity**						
Yes	129 (19.4)	286 (42.9)	251 (37.7)	8 (1.2)	323 (48.5)	335 (50.3)
No	211 (30.9)	286 (41.9)	186 (27.2)	56 (8.2)	455 (66.6)	172 (25.2)
	**χ^2^ = 29.253 *p* = 0.000**			**χ^2^ = 110.604 *p* = 0.000**		
**Social life**						
Yes	94 (20.2)	182 (39.1)	190 (40.8)	4 (0.9)	213 (45.7)	249 (53.4)
No	246 (27.9)	390 (44.2)	247 (28.0)	60 (6.8)	565 (64.0)	258 (29.2)
	**χ^2^ = 24.459 *p* = 0.000**			**χ^2^ = 87.918 *p* = 0.000**		
**Couples’ relationships**						
Yes	87 (21.8)	146 (36.6)	166 (41.6)	5 (1.3)	206 (51.6)	188 (47.1)
No	253 (26.6)	426 (44.8)	271 (28.6)	59 (6.2)	572 (60.2)	319 (33.6)
	**χ^2^ = 21.943 *p* = 0.000**			**χ^2^ = 31.847 *p* = 0.000**		
**Relationships with family members**						
Yes	62 (21.5)	109 (37.7)	118 (40.8)	2 (0.7)	133 (46.0)	154 (53.3)
No	278 (26.2)	463 (43.7)	319 (30.1)	62 (5.8)	645 (60.8)	353 (33.3)
	**χ^2^ = 12.037 *p* = 0.002**			**χ^2^ = 45.521 *p* = 0.000**		
**Relationships with friends**						
Yes	45 (21.6)	76 (36.5)	87 (41.9)	3 (1.4)	91 (43.8)	114 (54.8)
No	295 (25.9)	496 (43.5)	350 (30.7)	61 (5.3)	687 (60.2)	393 (34.5)
	**χ^2^ = 9.991 *p* = 0.007**			**χ^2^ = 33.329 *p* = 0.000**		

**Table 6 healthcare-10-00157-t006:** The Effect of Dysmenorrhea on University Performance (*n* = 1349).

Variable	Duration of Pain (%)			Intensity of Pain (%)		
	Few Hours	One Day	Two Days or More	Mild	Moderate	Severe
**Failing to focus on classes**						
Yes	181 (22.4)	327 (37.1)	300 (40.5)	14 (1.7)	419 (51.9)	375 (46.4)
No	159 (29.4)	245 (45.3)	137 (25.3)	50 (9.2)	359 (66.4)	132 (24.4)
	**χ^2^ = 21.993 *p* = 0.000**			**χ^2^ = 92.107 *p* = 0.000**		
**Individual study is affected**						
Yes	159 (20.3)	338(43.2)	285 (36.4)	20 (2.6)	412 (52.7)	350 (44.7)
No	181 (31.9)	234 (41.3)	152 (26.8)	44 (7.8)	366 (64.6)	157 (27.7)
	**χ^2^ = 27.237 *p* = 0.000**			**χ^2^ = 52.250 *p* = 0.000**		
**Volume of information would be smaller**						
Yes	112 (19.7)	232 (40.8)	225 (39.5)	8 (1.4)	276 (48.5)	285 (50.1)
No	228 (29.2)	340 (43.6)	212 (27.2)	56 (7.2)	502 (64.4)	222 (28.5)
	**χ^2^ = 28.038 *p* = 0.000**			**χ^2^ = 78.394 *p* = 0.000**		

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
