# Peer review of "The Prevalence, Management and Impact of Dysmenorrhea on Medical Students’ Lives—A Multicenter Study"

_healthcare, 2022, doi:10.3390/healthcare10010157_

Round 1

Reviewer 1 Report

Abstract

In the first sentence of the abstract, I would suggest at least one sentence of introduction to the subject, there are still a lot of free characters to use in the abstract

Research has already been done, it should be past tense, not present tense in the abstract.

The authors wrote that: „The aims of this study are to assess the prevalence of dysmenorrhea and to determine the impact of dysmenorrhea on the quality of life of medical students”  - therefore I think that the title of the publication should also contain the information about the prevalence of dysmenorrhea not only the impact of menstrual symptoms.

In the abstract, in the results and discussion section, there is a mix of words in the present and past tense, scientific papers (your results) should be wtitten in the past tense.

Introduction

I consider the introduction as a very poor, there is no enough medical and physiological information about dysmenorrhoea. There is a lack of basic descriptions of dysmenorrhea.

I propose to expand the introduction on the physiology and symptoms of dysmenorrhoea, and move to the discussion section of information that will clearly fit into the discussion of the results with existing literature.

Once again, the present tense was used for the purpose instead of the past tense.

Materials and Methods

Please attach a proprietary questionnaire to this manuscript

Results

I am wondering about the large age discrepancy in the study group and the time of the first menstruation.

Data from the literature on the subject say that a woman's age and the time of the first menstruation are important in relation to the occurrence and degree of symptoms of dysmenorrhea, so I would propose to consider a more homogeneous group, as there may be physiological differences in the perception of dysmenorrhea between women who are 18 years old and 30 years old, symptoms of the dysmenorrhea decrease with age.

I suggests removing figures and replacing them with a table.

Discussion

There is no need to inform the reader, for the third time, in the first sentence of the discussion section how many people participated in the study. In the discussion section, the reader should be more aware of which interpretations relate to the authors own research.

The manuscript have shown the readers also the methods of dealing with painful menstruation and there are also described, mayby the title of the manuscript should also be supplemented with this aspect.

Conclusions

Most of the sentences of the sections conclusion do not refer Authors research, it is only general opinion about painful menstruation. Conclusion should be rewritten.

Author Response

Thank you for the well welcomed suggestions. I Hope that this form of the article is according to your expectations.

Reviewer 2 Report

Thanks for giving me the opportunity of reviewing this article entitled: he impact of menstrual symptoms on medical students life – a multicenter study

The article is well written and provide a lot of data about menstrual symptoms that are very interesting.

However, a few improvements seem to be necessary to provide a more readable information about the results.

In the abstract please provide some numeric data that are more reliable for the reader and allow some interpretation.

Introduction:

There are many studies that does exists about prevalence of dysmenorrhea, (Armour et al., Jamieson et al., Pitts et al., Zondervan et al., Margueritte et al.). I think some data about prevalence should be wise in the introduction.

When you are talking about women avoiding exercising during menstruation, please provide some data about the study of Vlachou et al. that could give us some idea of the prevalence of theses women.

Material and methods:

There aren’t enough details about questions that are inside the questionnaire that is given to the women. The reader needs to know how they are formulated and if the answers: are binary (yes/ no), numerical, or whatever in order to explain your results and maybe to compare them to other study.

Please provide theses details that are mandatory for the discussion and for the interpretation of your results.

Statical analyses: does this study has been approved by an institutional review board or an ethic committee?

Results:

How much is your study population (population of interest), How do you manage your missing data?

A table where all the characteristics of your study population is missing. Please provide one.

On the other hand, some figure doesn’t seem to be necessary as you can provide these results within the text (for example figure 4 and 5)

In every figure or table please provide the number of your sample within the caption.

In your table 1, 2 and 3, it could be wise to present the results of the no pain category that could improve the understanding of your table. Also, the yes/no in the first column does it refer to the dysmenorrhea or to university activity, social life etc.. If so, I think these variables should be at the left side of the column.

Your table will be more understandable.

In your discussion you don’t mention the bias within this study:

-declarative bias as the women choose to answer what they want and what they feel.

-the size of your sample

Is your sample representative of your study population?

How do you manage your missing data if you have some or not?

Evaluation bias: we don’t know how you evaluate all the symptoms : Some evaluations are different if they are binary or within a scale with a threshold or whatever, please discuss this regarding to other study results .

Does this study provide some data that could be reproductible within other European countries?

The major strength of your study is that you have, I think a subsequent proportion of student given the fact that there is an online questionnaire and that your results seem to be reliable.

Author Response

Thank you for the welcomed suggestions. I Hope that this form of the article is according to your expectations.

Round 2

Reviewer 1 Report

I received the revised manuscript for review. I'm glad to see that the authors took reviewers' advice into consideration. The current manuscript is better than the one before it